# Metagenomic Changes of Gut Microbiota following Treatment of *Helicobacter pylori* Infection with a Simplified Low-Dose Quadruple Therapy with Bismuth or *Lactobacillus reuteri*

**DOI:** 10.3390/nu14142789

**Published:** 2022-07-06

**Authors:** Maria Pina Dore, Rosangela Sau, Caterina Niolu, Marcello Abbondio, Alessandro Tanca, Stefano Bibbò, Mariafrancesca Loria, Giovanni Mario Pes, Sergio Uzzau

**Affiliations:** 1Dipartimento di Medicina, Chirurgia e Farmacia, University of Sassari, Viale San Pietro 8, 07100 Sassari, Italy; caterina.niolu@hotmail.com (C.N.); mariafrancescaloria@libero.it (M.L.); gmpes@uniss.it (G.M.P.); 2Dipartimento di Scienze Biomediche, University of Sassari, Viale San Pietro 43B, 07100 Sassari, Italy; rosangela.sau@gmail.com (R.S.); mabbondio@uniss.it (M.A.); aletanca@uniss.it (A.T.); uzzau@uniss.it (S.U.); 3CEMAD Digestive Disease Center—Fondazione Policlinico Universitario Agostino Gemelli IRCCS, 00168 Rome, Italy; s.bibbo@gmail.com

**Keywords:** probiotics, randomized clinical trial, human gut microbiota, 16S rRNA gene sequencing

## Abstract

Background: Probiotic supplementation to antibiotic regimens against *Helicobacter pylori* infection has been proposed to improve eradication rate and to decrease detrimental effects on gut microbiota. Aims: To evaluate microbiota modifications due to a low-dose quadruple therapy with bismuth or *Lactobacillus reuteri*. Methods: Forty-six patients infected with *H. pylori* were prospectively enrolled in a single-centre, randomized controlled trial to receive b.i.d. with meals for 10 days low-dose quadruple therapy consisting of rabeprazole 20 mg and bismuth (two capsules of Pylera^®^ plus 250 mg each of tetracycline and metronidazole), or the same dose of rabeprazole and antibiotics plus Gastrus^®^ (*L. reuteri*), one tablet twice-a-day for 27 days. Stool samples were collected at the enrolment, at the end and 30–40 days after the treatment. Gut microbiota composition was investigated with 16S rRNA gene sequencing. Results: Eradication rate was by ITT 78% in both groups, and by PP analysis 85.7% and 95.5% for Gastrus^®^ and bismuth group, respectively. Alpha and beta diversity decreased at the end of treatment and was associated with a reduction of bacterial genera beneficial for gut homeostasis, which was rescued 30–40 days later in both groups, suggesting a similar impact of the two regimens in challenging bacterial community complexity. Conclusions: Low-dose bismuth quadruple therapy proved to be effective with lower costs and amount of antibiotics and bismuth. Gastrus^®^ might be an option for patients with contraindications to bismuth. *L. reuteri* was unable to significantly counteract dysbiosis induced by antibiotics. How to administer probiotics to prevent gut microbiota alterations remains an open question.

## 1. Introduction

*Helicobacter pylori* infection is common among populations and can cause devastating injury to human beings [1,2]. *H. pylori* infection is etiologically associated with gastritis, peptic disease, gastric cancer and mucosa-associated lymphoid tissue lymphoma [3]. For this reason, its diagnosis and treatment are strongly recommended [3]. However, only a small number of regimens reliably yield cure rates higher than 95%, due to *H. pylori* resistant strains, patients’ non-compliance and adverse events [4,5]. Side effects related to treatment may be the result of detrimental perturbations of the gut microbiota induced by antibiotics. The use of proton pump inhibitors (PPIs), an important component of anti-*H. pylori* regimens, also demonstrated to significantly change the gastric and intestinal microbial milieu [6]. To enhance efficacy of antimicrobial therapy against the bacterium, researchers moved considerable interest to using probiotics in addition to antibiotic regimens. For example, probiotics can reliably increase the cure rate ≥ 90% in regimens usually achieving cure rates of about 80% [7,8,9,10,11,12,13]. A meta-analysis including 5792 patients reported that probiotic supplementation improved the eradication rate by 10% (*p* < 0.00001) and significantly reduced side effects and gastrointestinal symptoms compared with controls; however, a significant between-study heterogeneity (*I*^2^ = 75%) was detected [13]. Similar results were reported in the meta-analysis performed by Shi et al. [10]. Probiotics given before, throughout and within the two weeks following the eradication treatment enhanced eradication rate, especially when combined with the bismuth quadruple regimen [10]. These results were further confirmed in 16 additional randomized controlled trials (RCTs) recruiting a total of 2466 patients [11]. The bismuth quadruple therapy supplemented with probiotic showed a statistically significant higher eradication rate compared to patients treated without probiotic supplementation (*p* = 0.0001) and *Lactobacillus* showed the highest eradication rate in comparison to other genera [11]. Also in children, *Lactobacillus* adjunct to triple therapy was able to increase *H. pylori* eradication rates and to significantly reduce the incidence of diarrhea related to therapy [9]. The adjunct of bismuth to amoxicillin, clarithromycin and PPIs determined a statistically significant reduction of alpha diversity and an increase in relative abundance of *Bifidobacterium adolescentis* and *Enterococcus faecium* in the gut microbiota [14]. In a prospective, randomized, double-blind, placebo controlled trial, supplementation of triple therapy with non-viable *L. reuteri* DSM17648 exerted a beneficial effect on the gut microbiota and reduced gastrointestinal symptoms, although not improving the eradication rate of *H. pylori* [15]. The esomeprazole, amoxicillin, furazolidone and bismuth b.i.d. regimen supplemented with *E. faecium* and *Bacillus subtilis* three times a day for four weeks, in a multicenter RCT, was associated with a faster taxa level restoration and increased Bacillus and Lactobacillales taxa [16]. Interestingly, an increase of Lactobacillus and Bifidobacterium has been reported to be also associated with *H. pylori* eradication after quadruple therapy containing bismuth [17].

In an open-label pilot study conducted in Sardinia, supplementation of Gastrus^®^ (*L. reuteri* DSM 17938 and *L. reuteri* ATCC PTA 6475) (BioGaia, Stockholm, Sweden), to the traditional quadruple therapy, was able to achieve an eradication of 84.8% and 79.6% according to per protocol (PP) and intention-to-treat (ITT) analysis, respectively [18].

In this study, we aimed (i) to confirm the efficacy of a low dose quadruple therapy containing bismuth versus *L. reuteri* on *H. pylori* eradication; (ii) to assess the compliance and the occurrence of adverse effects in patients treated with the two regimes; and (iii) to compare the impact of the two therapy protocols on the gut microbiota.

## 2. Materials and Methods

### 2.1. Study Design

Adult patients with *H. pylori* infection were randomized in a prospective open-label controlled-trial to receive a bismuth-based treatment or a probiotic-based treatment. Randomization was generated using the GraphPad software [19]. The treatment regimen was kept in a sealed envelope and opened before allocation of the patient. The study was conducted at the Department of Internal Medicine, Gastroenterology section, University of Sassari, Italy.

For each patient, three visits were scheduled. In the first visit (T0), the following information/samples were collected: (i) demographic data and anthropometric parameters, including body height, weight, as well as a detailed medical history and medications taken; (ii) a fecal sample; (iii) gastrointestinal symptoms, evaluated by using the Gastrointestinal Symptoms Rating Scale (GSRS) questionnaire, and treatment allocation (Figure 1).

In the second visit, after the end of the treatment (T1), the compliance and side effects were assessed and a second stool sample collected. In the final visit, 30–40 days after the end of treatment (T2), the last stool sample was collected, the efficacy of treatment checked and gastrointestinal symptoms re-evaluated (Figure 1).

### 2.2. Patient Eligibility

Patients older than 18 years referred to upper endoscopy or to a gastroenterology visit for dyspeptic symptoms, found positive for *H. pylori* infection and available to give stool samples, were invited to participate in the trial and to sign the informed consent.

### 2.3. Ethical Issues

The study was conducted in accordance with the Declaration of Helsinki, and approved by the Institutional Review Board of the local “Comitato Etico ASL No. 1 di Sassari” (Prot. No. 2358/CE). This study was registered on Clinical-Trials.gov, number NCT00669955. There was no pharmaceutical company influence in any phase of the study.

### 2.4. Exclusion Criteria

Presence of malignancy, pregnancy or lactation, clinically significant diseases, a recent gastroenteritis or bowel prep for colonoscopy were considered exclusion criteria. Patients who had a history of drug or alcohol abuse were also excluded. Allergy to any component of treatment regimens used in the trial or PPIs, anti-secretory drugs, antibiotics or probiotics taken one month preceding the enrollment were additional exclusion criteria.

### 2.5. H. pylori Status

Patients were defined as positive for *H. pylori* infection based on the detection of bacteria on gastric specimens and/or a positive ^13^C-Urea Breath Test (^13^C-UBT) (delta value greater than 5‰) and/or a positive stool antigen test. Eradication was confirmed by a negative ^13^C-UBT or a negative stool antigen test, both performed 30–40 days after treatment completion.

### 2.6. Medication

Both regimens included: rabeprazole 20 mg, tetracycline 500 mg and metronidazole 500 mg twice a day. One regimen was supplemented with bismuth subcitrate potassium 140 mg (Pylera^®^), 2 capsules twice a day, and the other one with Gastrus^®^ (BioGaia), 1 chewable tablet in the morning and in the afternoon. Both regimens were given with the midday and evening meals for 10 days with the exception of Gastrus^®^ tablets that were given for 27 days (Table 1).

### 2.7. Patients’ Compliance and Side Effects

At the time of enrollment, patients were instructed to return all drug containers after completion of therapy. More than 90% of the total recommended dose taken was considered excellent. Complained side effects were recorded after the end of treatment.

### 2.8. Characterization of the Gut Microbiota

Fecal samples were kept at 4 °C for a maximum of four hours after collection and stored at −80 °C until processed.

DNA was extracted from fecal samples with the QIAamp Fast DNA Stool Mini Kit (Qiagen, Hilden, Germany). DNA quantification was performed using a Qubit™ Fluorometer with the dsDNA High Sensitivity assay kit (Thermo Fisher Scientific, Waltham, MA, USA). Amplicon libraries were constructed using Illumina’s recommendations as previously described, amplifying the variable region 4 (V4) of the 16S rRNA gene [20]. Library quality control validation was performed through the TapeStation 4150 using the D1000 ScreenTape System (Agilent Technologies, Santa Clara, CA, USA). Subsequently, libraries (average size 425 bps) were quantified with a Qubit kit, normalized and pooled. DNA sequencing was performed in service using a MiSeq sequencer with the MiSeq Reagent Kit v2 (Illumina, San Diego, CA, USA), in paired-end modality and with 251 bases in each direction. The bioinformatic analysis of 16S rRNA gene sequencing data was performed as detailed elsewhere [20]. The Amplicon Sequence Variant (ASV) table was mapped against pre-formatted SILVA 138 SSURef NR99 database, using a 99% identity criterion.

In order to assess community richness and diversity, the ASV table was multiply rarefied to 90% of the minimum sample reads, and alpha and beta diversity were computed using phyloseq and visualized with ggplot2 (v.3.3.3) in R [21]. Specifically, alpha diversity and richness were calculated through Shannon’s index and number of observed ASVs, respectively. Beta diversity was calculated using weighted and unweighted UniFrac distances [22] and visualized by principal coordinates’ analysis (PCoA) plots.

### 2.9. Statistical Analysis

The eradication rates of *H. pylori* infection in the two groups were assessed by ITT, including all eligible patients enrolled in the study, and PP analysis, which excluded patients lost to follow-up, by using the Wald estimator. The significance threshold was set at *p* < 0.05. Ninety-five percent confidence intervals (95% CIs) were also calculated. Modification of symptoms after treatment in the two groups was categorized as: amelioration, disappearance, no modification and worsening, and compared.

The ASV data were aggregated based on the taxonomic assignment and subjected to differential statistical analysis between groups using the edgeR algorithm available on the Galaxy-P platform (v3.34.0 + galaxy1) [23,24,25], setting default parameters, except for trimmed mean of M-values as normalization [26]. In case of samples from the same patient analyzed at different times, the variable “patient” was added as additional factor and an additive linear model was used. Only patients with a documented *H. pylori* eradication were considered for the differential analysis. A correction for multiple tests was applied by calculating the False Discovery Rate (FDR) [27], setting FDR < 0.05 as a significant threshold. Moreover, only taxa present in at least 50% of the samples in a minimum of one group were considered as statistically significant. Differences in alpha diversity between groups were assessed by using the Wilcoxon rank sum test, setting a paired test in case of comparison between two different time points for the same patient. Significance testing between groups for beta diversity was assessed via permutational multivariate analysis of variance (PERMANOVA) using the vegan package (v.2.5-7) in R [28]. The heatmap illustrating the distribution of the ASVs belonging to *Lactobacillus* was created with the web application Morpheus (https://software.broadinstitute.org/morpheus/, accessed on 15 November 2021).

## 3. Results

### 3.1. Eradication Rate

Forty-six patients were randomized: 23 patients in the Gastrus^®^ group (mean age 57.2 ± 11.5 years; 14 women), and 23 in the bismuth group (mean age 51.2 ± 13.1 years; 16 women). The proportion of current and former smokers was similar in both groups (8.7% current smokers vs. 8.7%, and 28.6% former smokers vs. 30.4%, respectively); according to ITT analysis, the eradication rate was identical in both groups (78% vs. 78%), while, for PP, was 85.7% in the Gastrus^®^ group and 95.5% in the bismuth group, respectively (Table 2). Failure patients were treated with Pylera^®^ at the full dosage as indicated by the Pharma.

### 3.2. Compliance and Side Effects

Among the 40 patients who completed the trial, side effects including nausea, diarrhea, bloating, fatigue, bitter taste and dark stool, albeit not severe, were complained more frequently in the bismuth group compared with Gastrus^®^ group (26.3% vs. 9.5%; however, this difference was not statistically significant.

The compliance was kept higher than 90% in 93% and 91% of patients assigned to Gastrus^®^ and bismuth groups, respectively, within the whole treatment period.

The improvement and/or disappearance of gastrointestinal symptoms, recorded according to the GSRS questionnaire, was superior in the Gastrus^®^ group compared to bismuth group (68% vs. 53%), albeit not significantly.

### 3.3. Gut Microbiota Changes

Fifty-eight fecal samples were collected from 22 patients (9 from Gastrus^®^ group and 13 from bismuth group). Stool samples from patients lost to the follow up, or for whom the intervention was ineffective, as well as samples not passing the quality control, were excluded from the differential analysis. A total of 3,566,070 reads were obtained (61,484 on average per sample), corresponding to 2732 ASVs. These, in turn, were taxonomically assigned to 51 phyla, 93 families, and 212 genera, as described in Appendix A.

Alpha diversity and richness of the bacterial community, evaluated by the Shannon’s index and number of observed ASVs, respectively, did not differ between the two study groups before (T0), at the end of treatment (T1), and 30–40 days after the end of treatment (T2) (Appendix A).

In addition, differences among the three time points within the same group were investigated. In both Gastrus^®^ and bismuth groups, alpha diversity (Shannon’s index), and richness (number of observed ASVs) dropped at time T1 compared to baseline (Gastrus^®^, *p* = 0.002; bismuth, *p* = 0.0078) and partially recovered at T2 (Gastrus^®^, *p* = 0.002; bismuth, *p* = 0.0313) although to a lesser extent in the bismuth group.

Beta diversity between and within groups was also investigated through PCoA performed with ASV data. PCoA plots did not show significant difference between groups, according to both weighted and unweighted UniFrac (Appendix A). Differences were more pronounced when comparing the three time points within the same group, either with weighted or with unweighted UniFrac within Gastrus^®^ and bismuth groups (*p* = 0.001) (Appendix A).

A dramatic impact of the treatments on the gut microbiota composition was detected in both groups. Consistently with alpha-diversity dynamics, 30–40 days after treatment (T2), all gut bacterial members had recovered abundance levels similar to those measured at T0, with the sole exception of Barnesiella (in Gastrus^®^ group only). T1 was the only time point at which a few bacterial taxa showed a significantly different abundance between Gastrus^®^ group and bismuth group; specifically, Akkermansiaceae and the corresponding phylum Verrucomicrobiota, as well as Acidaminococcaceae, were more abundant in the bismuth group (Table 3).

Detailed lists of differential taxa are provided in Table 4 (at the genus level) and in Appendix A (at the phylum and family levels, respectively).

Several microbial taxa significantly changed in abundance following drug treatment (T1 vs. T0), e.g., Gram-negative Proteobacteria and Desulfobacterota, varied similarly in both treatment groups. Proteobacteria were detected in all samples at T0 and T1 and their relative abundance increased, as expected, after antibiotic treatment. Desulfobacterota, a former proteobacterial class, recently recognized as a novel phylum, were detected as well with a reduced relative abundance and in a lower number of samples (Appendix A). The decrease of Verrucomicrobiota in T1 was more evident in the Gastrus^®^ group (FDR < 0.000001) compared with bismuth group (FDR < 0.02); consistently, Akkermansiaceae and Akkermansia significantly changed only in the Gastrus^®^ group. No variation of Firmicutes was observed in any group, given the opposite behavior of families belonging to this phylum after antibiotic treatment (see below). Finally, a decrease of the Bacteroidota phylum was observed in the bismuth group at T1. Looking at genera, Enterococcus was found as more abundant at T1 in both groups. Furthermore, a decreased abundance of Akkermansia, Alistipes, Subdoligranulum, Coprococcus and Parabacteroides was detected in the Gastrus^®^ group, as well as an increase of Lactobacillus and Granulicatella. A decrease of taxa generally considered beneficial, such as Roseburia and Fusicatenibacter, was observed in the bismuth group.

Despite Gastrus^®^ supplementation given for 27 days (i.e., up to 17 days after antibiotic withdrawal), no significant differences were observed at T2 between the two groups.

Lactobacillus colonization from T0 to T2 was further explored by comparing the ASVs assortments and their relative abundances in both study groups (Appendix A). Samples from Gastrus^®^ group did not show colonization by a predominant Lactobacillus ASV, neither at T1 nor at T2. Clearly, different ASVs of Lactobacillus contributed to the taxon increase at T1 in both sample groups, suggesting the involvement of different Lactobacillus strains.

## 4. Discussion

In this RCT, the low-dose bismuth quadruple therapy given b.i.d. for 10 days achieved an eradication rate of 94.7% by PP analysis according to previous studies conducted in the same geographical area [18]. The cure rate decreased to 78% by ITT analysis. Replacement of bismuth with Gastrus^®^ proved to be less effective failing to cure nearly 10% of patients. As previously observed, these data highlight that *L. reuteri* DSM 17938 and *L. reuteri* ATCC PTA 6475 were not able to completely replace the bismuth action against *H. pylori* [18]. The choice of *L. reuteri* in substitution of bismuth was based on the evidence that this bacterium produces reuterin (3-hydroxypropionaldehyde), an antimicrobial compound against *H. pylori* [29]. Moreover, *L. reuteri* hampers *H. pylori* colonization of human gastric mucosa through the inhibition of *H. pylori* binding to glycolipid receptors [30]. Yao et al., in an elegant study published last year, elucidated the underlying mechanism of bismuth action against *H. pylori* [31]. Functional analysis showed that bismuth downregulates CagA and VacA virulence proteins, impairs antioxidant response and compromises purine, pyrimidine, amino acid and carbon metabolic pathways involved in bacterial growth. Moreover, bismuth inhibits *H. pylori* colonization breaking off the assembly of the bacterial flagellum [31]. These findings may explain the superiority of bismuth with respect to *L. reuteri* in terms of efficacy against *H. pylori*.

In addition, low-dose bismuth resulted in being highly effective, as an empirical regimen, similarly to high-dose bismuth quadruple therapy. For example, in 2100 European patients treated with Pylera^®^ (single-capsule bismuth quadruple therapy), according to the Pharma instructions (i.e., three capsules every six h for 10 days), eradication rate was greater than 90% in clinical practice. However, 60 patients complained of severe adverse events, with 1.7% of them requiring to stop the treatment [32]. In our study, patients were exposed to a regimen containing a low dose quadruple therapy and avoided almost 10 gr of bismuth, 5 gr of metronidazole and 5 gr of tetracycline and severe side effects were not experienced. However, the Gastrus^®^ group complained fewer side effects compared to the bismuth group, as previously reported [18]. Moreover, a greater percentage of patients claimed gastrointestinal symptoms improvement and/or disappearance in the Gastrus^®^ group compared to the bismuth group. Interestingly, in both study groups, the eradication rate by ITT analysis was identical.

The 16S metagenomic sequencing revealed that alpha diversity significantly changed soon after eradication compared with baseline. Alpha diversity is an ecological, cumulative measure of the complexity of the microbial community, used as a hallmark of homeostasis and resilience of the gut microbiota, being inversely correlated with mucosal inflammation and permeability of the intestinal barrier [33]. Treatment-dependent disruption of the gut microbiota, highlighted by the drop of alpha diversity at T1, showed a recovery 30–40 days after treatment, albeit not complete. The beta diversity followed a similar pattern, confirming a possible restoration at the end of intervention in each study group.

Given that weighted UniFrac accounts for the abundance of the observed microorganisms (number of reads assigned to an ASV), this result may suggest that the two regimens, either supplemented with *L. reuteri* strains or with bismuth, are both somehow able to shape biodiversity.

While the dynamics of alpha diversity were similar, some key members of the gut microbiota showed Gastrus^®^- or bismuth-specific changes of their relative abundance. Both treatments were associated with an increase of Proteobacteria, a large phylum of Gram-negative bacteria, mostly facultative aerobes/anaerobes, including several pathobionts. Members of this phylum were not detected as differentially abundant at family or genus levels, suggesting that most Proteobacteria members were evenly modified by treatment, with the exception of Morganellaceae that showed a significant increase of its relative abundance only in the Gastrus^®^ group. It is worth noting that opportunistic genera belong to this family and their increase in antibiotic-dysrupted microbiota has already been reported. Similarly, Bandy and Zhang reported an increased opportunistic genus in (Morganella, Proteus, and Providencia) dysbiosis due to antibiotics, indicating that bismuth may be more effective in controlling the opportunistic blooming of Morganellaceae during antibiotic treatment [34,35]. Recently, a large group of Proteobacteria members were reclassified in the novel phylum Desulfobacteroidota, based on phylogeny and metabolic functions [36]. In the present study, we observed for the first time an opposite behavior of Proteobacteria and Desulfobacteroidota under antibiotic treatment, highlighting a different response of the two phyla to gut environmental changes.

According to a previous study aimed at evaluating the short- and long-term effect of bismuth quadruple therapy on the gut microbiota in a pediatric population allergic to penicillin [37], we observed a significant decrease of the Bacteroidota phylum, including the butyrate-producer Butyricimonas, after bismuth treatment. Instead, Alistipes and Parabacteroides were significantly reduced in the Gastrus^®^ group. In addition, a decrease of other specific Bacteroidota members, including Bacteroides, Barnesiella and Odoribacter, was observed in both treatment groups. Although Bacteroidota members showed different susceptibility to bismuth, given the cumulative effect at phylum level, the overall response to bismuth treatment was higher.

Opposite trends were observed for different genera belonging to Firmicutes, determining, in turn, a lack of the overall phylum variation. In the bismuth group, there was a significant increase of Enterococcus, Granulicatella and Lactobacillus. The aerobic pathobiont Enterococcus, together with Proteobacteria members, represent a hallmark of gut microbiota dysbiosis. Granulicatella, a nutritional variant Streptococcus, has been reported to increase after *H. pylori* eradication and to contribute to the progression of precancerous gastric lesions to cancer [38]; additionally, it might potentially cause endocarditis [39]. Studies on *Lactobacillus* modification in the gut microbiota determined by bismuth quadruple therapy with or without probiotics reported contrasting results [17,40,41]. In our study, bismuth caused a significantly increased *Lactobacillus* at the end of treatment, while Gastrus^®^ supplementation did not.

In a recent study conducted in Taiwan, fecal samples obtained at the end of therapy from patients treated with bismuth quadruple therapy (q.i.d. for 14 days) displayed a decreased abundance of Verrucomicrobia and Bacteroidota, similarly to our findings [42]. The same regimen given for 10 days, six weeks after *H. pylori* eradication, caused a statistically significant decrease of *Akkermansia*, *Bifidobacterium*, *B. fragilis* and *Faecalibacterium*
*prausnitzii* [43]. *Akkermansia* spp. are recognized as beneficial members of the gut microbiota, capable of growing on mucin as a sole source of nutrients and are generally associated with a healthy gut mucosa [44]. It is worth noting that we observed a less pronounced reduction of Verrucomicrobia and the correlated family Akkermansiaceae in the bismuth group.

Interestingly, Barnesiella showed a dramatic reduction of its relative abundance even 30–40 days after treatment with the Gastrus^®^ regimen, with a fold change almost identical to that observed immediately after treatment. Barnesiella is an important genus for the gut mucosal health, especially when the microbiota is challenged by antibiotics, as it is reported to be associated with pathobionts’ eradication [45] and to reduce susceptibility to colitis induction in mice [46].

Although in the literature there are studies demonstrating that probiotic supplementation may reduce the detrimental effects of antibiotics in the gut microbiota [47,48], in our study, abundance increment or reduction did not dramatically differ in the probiotic group compared to the bismuth group.

A number of limitations in our study need to be mentioned. First, the study design was open-labelled, which may have hampered minimizing the researcher’s bias as well as the placebo effect. However, although this may have impacted eradication rate, it may have hardly influenced the gut microbiota changes. The relatively small size of treatment groups may have theoretically reduced the statistical power, although in a previous study [18], enrolling a total of 99 patients, similar eradication rates were obtained by using the same two treatment regimens. Moreover, stool samples from non-responders were not sequenced and the number of stool samples failing quality control was not negligible (14 out of 36 samples). Although detailed instructions on stool collection were given to the patients, after an accurate control, we were forced to exclude several samples because they were stored at room temperature, or for more than 2 h, and this may have affected results.

## 5. Conclusions

Overall, our findings prove that twice-a-day low-dose bismuth quadruple therapy given for 10 days is able to achieve high eradication rates with lower costs, amount of antibiotics and bismuth. However, a proper non-inferiority study against high doses would be required to confirm that both are equally effective.

Replacement of bismuth with Gastrus^®^ might be an option for patients with contraindications to bismuth. However, *L. reuteri* in the method of administration used in this study was not able to significantly counteract alterations induced in the gut microbiota by antibiotics, still leaving open the question on how to administer probiotics (dosage, timing, for how long, which strain, or a combination of strains) to prevent dysbiosis.

## Figures and Tables

**Figure 1 nutrients-14-02789-f001:**
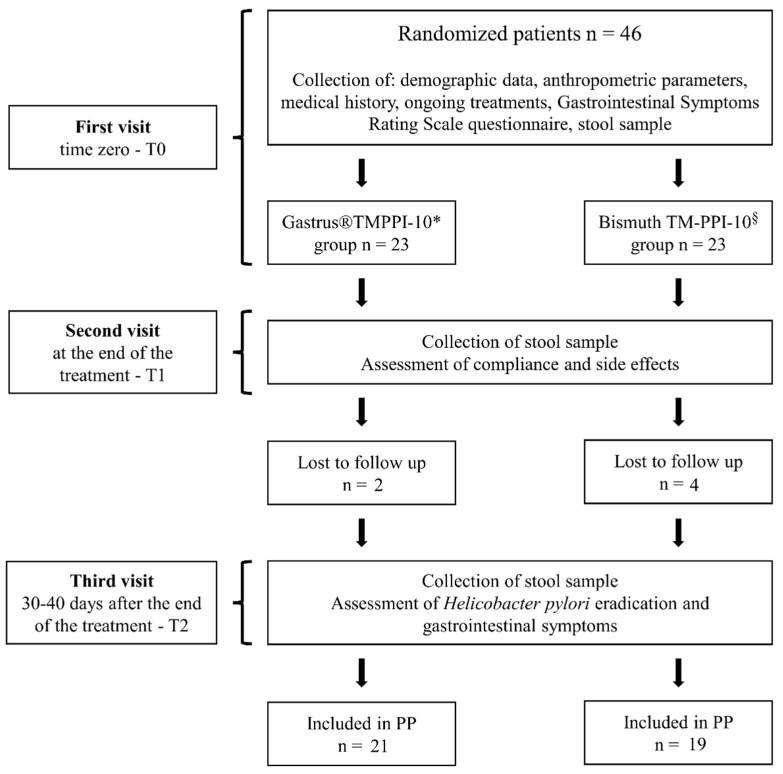
Study design of the controlled trial. The symbols * and § represent the treatment according to the Table 1.

**Table 1 nutrients-14-02789-t001:** Regimens used in the trial.

Gastrus^®^ Group	Bismuth Group
Rabeprazole 20 mg 1 cp	Rabeprazole 20 mg 1 cp
Metronidazole 250 mg 2 cp	Metronidazole 250 mg 1 cp
Tetracycline 250 mg 2 cp	Tetracycline 250 mg 1 cp
(b.i.d.: midday and evening meals)	Pylera^®^ 140 mg/125 mg/125 mg 2 cp
for 10 days	(b.i.d.: midday and evening meals)
Gastrus^®^ 1 tablet in the morning and 1 tablet in the afternoon for 27 days	for 10 days

**Table 2 nutrients-14-02789-t002:** Intervention status of 46 patients enrolled in the trial.

Status	Gastrus^®^ Group ^1^	Bismuth Group ^1^
Received intervention	23	23
Lost to follow-up	2	4
Completed trial	21	19
Intervention ineffective	3	1
Cure Rate ITT ^2^	78.3% (18/23)	78.3% (18/23)
95% CI ^4^-ITT	61.4–95.1	61.4–95.1
Cure Rate PP ^3^	85.7% (18/21)	94.7% (18/19)
95% CI ^4^-PP	71.4–100	85.6–100

^1^ Treatment regimens are specified in Table 1. ^2^ ITT: intention-to-treat analysis; ^3^ PP: per protocol analysis; ^4^ CI: Confidence Interval.

**Table 3 nutrients-14-02789-t003:** List of differential abundant taxa between Gastrus^®^ group and Bismuth group. Taxa with statistically significant difference (FDR < 0.05) identified in at least 50% of the samples in at least one group.

Taxonomic Level	Taxa	Time T1	
		logFC (Gastrus^®^/Bismuth) ^1^	FDR ^1^
Phylum	Verrucomicrobiota	−7.24	0.02329
Family	Akkermansiaceae	−9.30	0.02715
	Acidaminococcaceae	−11.14	0.02182

^1^ FC: Fold Changes; FDR: False Discovery Rate.

**Table 4 nutrients-14-02789-t004:** List of differential abundant genera when comparing T0 vs. T1 and T2. Genera with statistically significance (FDR < 0.05), identified in at least 50% of the samples in at least one group.

Genus	Gastrus^®^ Group				Bismuth Group			
	T1 < T0		T2 < T0		T1 < T0		T2 < T0	
	logFC (T1/T0)	FDR	logFC (T2/T0)	FDR	logFC (T1/T0)	FDR	logFC (T2/T0)	FDR
*Adlercreutzia*					−6.60	0.03803		
*Akkermansia*	−11.37	0.00730						
*Alistipes*	−12.18	0.00548						
*Anaerostipes*					−5.74	0.02256		
*Barnesiella*	−13.47	0.00280	−12.18	0.01277	−6.94	0.00996		
*Bilophila*					−8.20	0.00962		
*Butyricimonas*					−8.33	0.00962		
*Collinsella*					−8.05	0.00960		
*Coprococcus*	−9.70	0.04882						
*Flavonifractor*					−7.23	0.04680		
*Fusicatenibacter*					−8.80	0.00102		
*Intestinimonas*					−7.42	0.00962		
*Lachnoclostridium*					−7.06	0.00962		
*Megasphaera*					−10.41	0.00962		
*Monoglobus*	−11.23	0.00548			−8.52	0.00167		
*Odoribacter*	−10.65	0.03474			−9.56	0.00088		
*Oscillibacter*					−7.38	0.03803		
*Parabacteroides*	−9.44	0.04882						
*Parasutterella*					−6.63	0.00996		
*Romboutsia*					−7.80	0.00962		
*Roseburia*					−9.08	0.00167		
*Ruminococcus*	−8.07	0.04882			−9.23	0.00167		
*Subdoligranulum*	−11.22	0.03586						
**Genus**	**T1 > T0**		**T2 > T0**		**T1 > T0**		**T2 > T0**	
	**logFC (T1/T0)**	**FDR**	**logFC (T2/T0)**	**FDR**	**logFC (T1/T0)**	**FDR**	**logFC (T2/T0)**	**FDR**
*Enterococcus*	10.98	0.00548			12.81	0.00276		
*Granulicatella*					5.56	0.04680		
*Lactobacillus*					7.32	0.04909		

## Data Availability

The datasets presented in this study can be found in online repositories. The names of the repository/repositories and accession number can be found at: https://www.ebi.ac.uk/ena, PRJEB35028.

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
