# Peer review of "Metagenomic Changes of Gut Microbiota following Treatment of Helicobacter pylori Infection with a Simplified Low-Dose Quadruple Therapy with Bismuth or Lactobacillus reuteri"

_nutrients, 2022, doi:10.3390/nu14142789_

Round 1

Reviewer 1 Report

Comments to author(s):

Paper review “Metagenomic changes of gut microbiota following treatment of Helicobacter pylori infection with a simplified low-dose quadruple therapy with bismuth or Lactobacillus reuteri”.

The paper is well written, and the project involved a lot of work. The presentation of the context is very good.

Concerns raised include:

1.    Introduction

Lines 74-75, it is better to include the specific objectives of the study.

2.    Materials and Methods

2.1   Study design

Line 82, to include the name of the method.

In text citation for an online source: include the author and year of publication. The URL goes in the corresponding reference list entry.

2.2 Patient eligibility

Lines 99-101, the ethics approval information should be included.

2.5. Medication

Table 1, top line and bottom line to be added.

2.8 Statistical analysis

Line 158, in text citation, same as 2.1

3.    Results

Figure 2, box plots can be replaced by a paragraph.

4.    Discussion

Line 335, it is better to include the topic of your previous study e.g. According to a previous study on ….

Are there any limitations in your study?

5.    Other comments

Please read the paper carefully with regard to correct English

Author Response

Paper review “Metagenomic changes of gut microbiota following treatment of Helicobacter pylori infection with a simplified low-dose quadruple therapy with bismuth or Lactobacillus reuteri”.

The paper is well written, and the project involved a lot of work. The presentation of the context is very good.

Response: We thank the reviewer for his/her appreciation.

Concerns raised include:

  1. Introduction

Lines 74-75, it is better to include the specific objectives of the study.

Response: In the revised manuscript we added the following lines: “In this study we aimed i) to confirm the efficacy of a low dose quadruple therapy containing bismuth versus L. reuteri on H. pylori eradication; ii) to assess the compliance and the occurrence of adverse effects in patients treated with the two regimens; and iii) to compare the impact of the two therapy protocols on gut microbiotaË®. Page 2, lines 74-78.

  1. Materials and Methods

2.1   Study design

Line 82, to include the name of the method. In text citation for an online source: include the author and year of publication. The URL goes in the corresponding reference list entry.

Response: Thanks for the suggestion. The name of the software was added (GraphPad) and the URL moved to the reference list (Ref no. 19)

2.2 Patient eligibility

Lines 99-101, the ethics approval information should be included.

Response: Sorry, but we used the Nutrients Journal Template where the Institutional Review Board Statement and the Informed Consent Statement both go at the end of the manuscript.

However, according to your suggestion, the ethics approval information was added in a specific sub-section “2.3 Ethical issues

2.5. Medication

Table 1, top line and bottom line to be added.

Response: we did.

2.8 Statistical analysis

Line 158, in text citation, same as 2.1

Response: The URL was added as well as the corresponding reference (23).

  1. Results

Figure 2, box plots can be replaced by a paragraph.

Response: In the revised version the box plots in Figure 2 were replaced by a short text as follows: “In both Gastrus® and bismuth groups, alpha diversity (Shannon’s index), and richness (number of observed ASVs) dropped at time T1 compared to baseline (bismuth, P = 0.0078; Gastrus, P=0.002) and partially recovered at T2 (bismuth, P = 0.0313; Gastrus, p=0.002) although to a lesser extent in the bismuth group”. (page 6, lines 222-225)

  1. Discussion

Line 335, it is better to include the topic of your previous study e.g. According to a previous study on ….

Response: Done: page 10, lines 350-351.

Are there any limitations in your study?

Response: A new section concerning the study limitations has been added to the revised manuscript. Page 11, lines 389-397.

  1. Other comments

Please read the paper carefully with regard to correct English

Response: A text-editing service was asked for correction.

Reviewer 2 Report

INTRODUCTION

Lines 45-50: It would be worth commenting said meta-analyses also found probiotics reduced side effects of eradication therapies, but significant heterogeneity was observed among studies.

METHODS

Lines 131-148:

It is difficult to uniquely identify lactobacillus genus based on short 16S reads such as V4. Please state percent identity used for ASV clustering in Methods section. Also state confidence of ASV genus assignment on figure S5.

Please indicate full name of ASV acronym on first occurrence (Amplicon Sequence Variant). We may be familiar with it, but not every reader will.

Lines 150-155:

Please state statistical test used to compare eradication rate.

RESULTS

Figure 1 indicates 19 patients completed the study in Gastrus group and 21 in Bismuth, but Table 2 reports inverse numbers. Please clarify.

Lines 190 and 197: Clearly state that “this difference was not statistically significant” in both lines.

Lines 200-205: Excluding samples of patients not responding to H.pylori erradication is unjustified, since the aim of the study is to see effect on colonic microbiota. Antibiotics will kill colonic bacteria regardless of whether there is H.pylori in the stomach or not. Moreover, of 36 patients completing the study and with successful eradication, only 22 are included in metagenomic analysis, meaning 14 of 36 samples failed quality controls (38.9%). This is a very high number and casts doubts about microbiota sampling procedure. Please repeat metagenomic analyses by including the samples of the 4 subjects whose H.pylori was not eradicated, and provide detailed explanation in Discussion about why almost 40% of stool samples failed quality control.

DISCUSSION

Please add a paragraph discussing study limitations, including high number of stool samples failing quality control, open-label design of the study and small sample size.

CONCLUSIONS

Although low dose bismuth seems interesting, a head-to-head study against high dose is required to confirm both are equally effective. Conclusions should indicate such a study is warranted to confirm the interesting findings of this paper.

SUPPLEMENTARY MATERIALS

Please relabel groups A and B to Gastrus and Bismuth in all figures, to facilitate reading.

Author Response

INTRODUCTION

Lines 45-50: It would be worth commenting said meta-analyses also found probiotics reduced side effects of eradication therapies, but significant heterogeneity was observed among studies.

Response: In the revised manuscript the following sentence was added: “… the probiotic supplementation improved the eradication rate by 10% (p<0.00001) and reduced side effects of eradication therapy, but significant heterogeneity was observed between studies (I2 = 75%)”. Page 2, lines 46-49.

METHODS

Lines 131-148:

It is difficult to uniquely identify lactobacillus genus based on short 16S reads such as V4. Please state percent identity used for ASV clustering in Methods section. Also state confidence of ASV genus assignment on figure S5.

Response: Thanks for the comment.

We have mistakenly indicated a wrong reference with regards to the bioinformatic analysis applied, you can find the correct one in the revised manuscript. As reported elsewhere, fastq files obtained by the Miseq sequencer were subjected to denoising using DADA2 algorithm via QIIME2 pipeline. As a result, we produced an Amplicon Sequence Variant table, which is a higher-resolution analogue of the traditional OTU table. ASVs are inferred by a de novo process and can accurately reflect biological reality, because of single-nucleotide differences over the sequenced gene region. Taxonomic composition was then explored using pre-formatted SILVA 138 as the reference database.

The following sentence was added: “The Amplicon Sequence Variant (ASV) table was mapped against pre-formatted SILVA 138 SSURef NR99 database, using a 99% identity criterion.” (REF) Page 5, lines 154-156.

Please indicate full name of ASV acronym on first occurrence (Amplicon Sequence Variant). We may be familiar with it, but not every reader will.

Response: Thanks for the suggestion. Done, Page 5, line 154

Lines 150-155:

Please state statistical test used to compare eradication rate.

Response: Done

RESULTS

Figure 1 indicates 19 patients completed the study in Gastrus group and 21 in Bismuth, but Table 2 reports inverse numbers. Please clarify.

Response: Thanks for your advice. In the revised version of the manuscript, the mistake was corrected.

Lines 190 and 197: Clearly state that “this difference was not statistically significant” in both lines.

Response: We did

Lines 200-205: Excluding samples of patients not responding to H.pylori erradication is unjustified, since the aim of the study is to see effect on colonic microbiota. Antibiotics will kill colonic bacteria regardless of whether there is H.pylori in the stomach or not. Moreover, of 36 patients completing the study and with successful eradication, only 22 are included in metagenomic analysis, meaning 14 of 36 samples failed quality controls (38.9%). This is a very high number and casts doubts about microbiota sampling procedure. Please repeat metagenomic analyses by including the samples of the 4 subjects whose H.pylori was not eradicated, and provide detailed explanation in Discussion about why almost 40% of stool samples failed quality control.

Response: As stated in the revised manuscript (lines 74-78), the aim of the study was to "(i) confirm the efficacy of a low dose quadruple therapy containing bismuth or versus L. reuteri for on H. pylori eradication.... and (iii) to compare the impact of the two therapy protocols on gut microbiota". H. pylori, per se, is associated to changes of the intestinal microbiota and its presence might bias the comparison within and between the study groups. Finally, we acknowledge that 14 out of 36 samples not passing quality control is a very high number. Patients received clear instructions on how to store properly their samples (refrigerated for a maximum of 4 hours after collection). Unfortunately, after a more detailed verification we observed that for 14 samples instructions were not followed as recommended (for instance stored at room temperature, for more than 2 hours,and more). All factors able to sufficiently affect the relative abundance of bacterial data.

DISCUSSION

Please add a paragraph discussing study limitations, including high number of stool samples failing quality control, open-label design of the study and small sample size.

Response: In the revised manuscript we added the following paragraph: “A number of limitations in our study need to be mentioned. First, the study design was open-labelled which may have hampered minimizing the researcher’s bias as well as the placebo effect. However, although this may have impacted eradication rate, it may have hardly influenced the gut microbiota changes. The relatively small size of treatment groups may have theoretically reduced the statistical power, although in a previous study (ref n. 18) enrolling a total of 99 patients, similar eradication rates were obtained by using the same two treatment regimens. Finally, the number of stool samples failing quality control in our study was not entirely negligible and may have affected data consistency, albeit moderately” Page 2, lines 392-400

CONCLUSIONS

Although low dose bismuth seems interesting, a head-to-head study against high dose is required to confirm both are equally effective. Conclusions should indicate such a study is warranted to confirm the interesting findings of this paper.

Response: This is an interesting point. Multiple antibiotic regimens have been evaluated for H. pylori eradication. However, few regimens have consistently achieved high eradication rates. The major limitation in guiding therapy is the lack of available H. pylori antibiotic resistance rates.

An ideal therapy should have the following characteristics: lowcost, high rate of treatment success, simple to administer, well tolerated, and with few side effects.  While no therapy currently in use meets all these criteria, there are a number of effective, well-tolerated therapies available. Among these, bismuth quadruple therapy (Pylera) is currently the most recommended as a rescue therapy and in naïve patients. However, Pylera is difficult to find in Italy, and as specified in  the discussion contains a high concentration of antibiotics and bismuth.

Bismuth salts have been used to treat ‘gastritis’ and peptic ulcers for over 200 years, as well as syphilis before modern antibiotics became available or to prevent E. coli traveler’s diarrhea. Bismuth is directly bactericidal to H. pylori leading to bacterial lysis within 2 hours of ingestion. More importantly H. pylori does not develop resistance to the various bismuth salts which make bismuth an important part of H. pylori therapy for years to come.    

Although it has been reported that bismuth has an excellent safety record with the most problematic side effect being temporary discoloration of the tongue and development of black stools, there is a hypothetical concern for systemic absorption of bismuth leading to bismuth toxicity. In fact, we observed some cases of patients who required hospitalization in the intensive care because of toxicity by Pylera.  

We agree with the reviewer that a head-to-head study (high dose versus low dose) would be theoretically the best way to confirm if both are equally effective against H. pylori. However in our opinion this will expose patients to a potentially dangerous therapy, unethical per se, moreover bismuth is not any longer available on the market alone nor in a combined form (Pylera) and finally, the robust literature available on Pylera can be a good surrogate of a high-dose arm.

SUPPLEMENTARY MATERIALS

Please relabel groups A and B to Gastrus and Bismuth in all figures, to facilitate reading.

Response: Files in the supplementary material were modified accordingly.

Reviewer 3 Report

Manuscript nutrients-1782924 deals with the metagenomic changes of gut microbiota after treatment of Helicobacter pylori infection  using a low-dose therapy with bismuth or probiotics on the basis of Lactobacillus reuteri.

The topic is of interest given that Helicobacter pylori affects many people around the world. In addition, the use of probiotics seems to cure the symptoms caused by Helicobacter pylori. The authors conducted well a clinical trial using a certain number of patients (46). In these studies, however, the number of patients used must be defined by Power Analysis. Therefore, the authors must carry out Power Analysis to define the least sample size that can give clear results.

In addition, in the Introduction section it must be defined that if Helicobacter pylori symptoms are not cured, this disease may lead also to cancer. The authors in this statement must use a reference.

As far as the organization of the manuscript and the English language both are acceptable and of good level.

Some other comments within the text are as follows:

-Abstract

Line 17. ‘’ infected with Helicobacter pylori ‘’.

-Introduction

See above comments.

-Statistical analysis section and sample design

See the comment for Power Analysis.

-Outcomes and future perspectives

The study has a positive effect on the construction of future studies and the conclusions drawn by the reported results will help the relevant literature.

Given the very good technical quality of the paper, I suggest a minor revision prior to further consideration for publication.

Author Response

Comments and Suggestions for Authors

Manuscript nutrients-1782924 deals with the metagenomic changes of gut microbiota after treatment of Helicobacter pylori infection using a low-dose therapy with bismuth or probiotics on the basis of Lactobacillus reuteri.

The topic is of interest given that Helicobacter pylori affects many people around the world. In addition, the use of probiotics seems to cure the symptoms caused by Helicobacter pylori. The authors conducted well a clinical trial using a certain number of patients (46). In these studies, however, the number of patients used must be defined by Power Analysis. Therefore, the authors must carry out Power Analysis to define the least sample size that can give clear results.

Response: This population study was an additional group of a previous study (Ref. n 18). The aim of the previous study was to compare the efficacy against H. pylori by using the two regimens protocols used in the present study. The previous study was a prospective single-center, randomized, open-label trial conducted as two pilot studies in 99 adult patients with H. pylori infection, and the sample size was estimated based on noninferiority trial design. The estimated sample size was obtained with an online calculator (http://www.graphpad.com/quickcalcs/randomize2/). In the present study the number of patients used was not defined by Power Analysis because the major outcome was to analyze the metagenomic changes of gut microbiota after treatment of H. pylori infection using a low-dose therapy with bismuth or probiotics. However, although the sample size is relatively small, making the results underpowered about treatment efficacy, findings obtained in the present study are pretty similar to the previous one. Moreover, the present study aimed just to confirm previous results. The issue about the small sample size was added in the limitations. Page 11, lines 395-398.

In addition, in the Introduction section it must be defined that if Helicobacter pylori symptoms are not cured, this disease may lead also to cancer. The authors in this statement must use a reference.

Response: Actually, almost more than half of patients with H. pylori infection do not have symptoms and they may or may not develop gastroduodenal disease by the time. Among those with H. pylori infection, the lifetime risk of peptic ulcer is approximately 17% (1 in 6), and the lifetime risk of developing gastric cancer varies from approximately 0.6% to 22% worldwide (Ref. n 3)

On the other hand, H. pylori eradication is not always associated with disappearance of symptoms in patients with dyspepsia (NNT 17 for H. pylori eradication versus placebo) (Cochrane Database Syst Rev 2003;1:CD0020969). However, is universally recognized that the infection is invariably related to H. pylori gastritis and is etiologically associated to peptic ulcer and gastric cancer. For these reasons the gastroenterology community formally recognized H. pylori gastritis as an infectious disease and recommended that whenever H. pylori infection is diagnosed, it should be eradicated, and in more recent consensus statements a proactive approach to testing and treatment of H. pylori was recommended in family members of individuals diagnosed with active infection as well as high-risk local populations such as immigrants from high-risk countries.

The importance to eradicate H. pylori infection (because it may cause cancer)  is summarized in the revised manuscript in the introduction section. Page 1, lines 36-37.

As far as the organization of the manuscript and the English language both are acceptable and of good level.

Some other comments within the text are as follows:

-Abstract

Line 17. ‘’ infected with Helicobacter pylori ‘’.

Response: Thanks for the suggestion, we made the correction.

-Introduction

See above comments.

-Statistical analysis section and sample design

See the comment for Power Analysis.

-Outcomes and future perspectives

The study has a positive effect on the construction of future studies and the conclusions drawn by the reported results will help the relevant literature.

Given the very good technical quality of the paper, I suggest a minor revision prior to further consideration for publication.

Round 2

Reviewer 2 Report

Manuscript has been markedly improved, and I support it’s publication. However, a few minor concerns should be addressed first.

METHODS

Lines 164-169: Statistical test used to compare eradication rate still not mentioned, despite my previous request. Please, state whether you used a Chi-squared attest, and Armitage test, a T-Test, a Mann-Whitney test or whichever applies.

RESULTS

Lines 214-220: I can’t find solid evidence of H.pylori resulting in significant microbiota changes per se. Moreover, any changes likely to be small compared to the effect of antibiotic, PPI and bismuth therapy. Preferably, H.pylori-infected individuals should have been included so that beta-diversity analyses can pick them out and identify them as being different. Since this was not done, I’d recommend the authors to mention the fact that non-responders were not sequenced among limitations in Discussion section

DISCUSSION

I really beg you pardon, but saying that “the number of stool samples failing quality control was not entirely negligible” seems like an understatement. I’m all for publishing your paper, but please be up front and state that a significant number of samples (39%) did not meet quality criteria, and explain what happened (e.g. patients storing samples at room temperature). I totally understand this was not your fault, but please be transparent about it.

CONCLUSIONS

Although low dose bismuth seems interesting, a proper non-inferiority study against high dose would be required to confirm both are equally effective. Conclusions should indicate such a study is warranted to confirm the interesting findings of this paper.

Author Response

Reviewer #2

The manuscript has been markedly improved, and I support it’s publication. However, a few minor concerns should be addressed first.

METHODS

Lines 164-169: Statistical test used to compare eradication rate still not mentioned, despite my previous request. Please, state whether you used a Chi-squared attest, and Armitage test, a T-Test, a Mann-Whitney test or whichever applies.

Response: Sorry we did not understand the question: the statistical test is now specified. Page 5, line 166.

RESULTS

Lines 214-220: I can’t find solid evidence of H. pylori resulting in significant microbiota changes per se. Moreover, any changes likely to be small compared to the effect of antibiotic, PPI and bismuth therapy. Preferably, H.pylori-infected individuals should have been included so that beta-diversity analyses can pick them out and identify them as being different. Since this was not done, I’d recommend the authors to mention the fact that non-responders were not sequenced among limitations in Discussion section.

Response: Done, page 11 lines 399-400 and 404

DISCUSSION

I really beg you pardon, but saying that “the number of stool samples failing quality control was not entirely negligible” seems like an understatement. I’m all for publishing your paper, but please be up front and state that a significant number of samples (39%) did not meet quality criteria, and explain what happened (e.g. patients storing samples at room temperature). I totally understand this was not your fault, but please be transparent about it.

Response: we tried to change the sentence in order to avoid minimizing the significant number of stool samples failing quality control.

CONCLUSIONS

Although low dose bismuth seems interesting, a proper non-inferiority study against high dose would be required to confirm both are equally effective. Conclusions should indicate such a study is warranted to confirm the interesting findings of this paper.

Response: In the conclusions a sentence about that was added, page 11 lines 409-410

Sincerely thanks for your interest on our manuscript

Best regards

Maria Pina Dore